# Distribution, Vertical Transmission, and Cooperative Mechanisms of Obligate Symbiotic Bacteria in the Leafhopper *Maiestas dorsalis* (Hemiptera, Cicadellidea)

**DOI:** 10.3390/insects14080710

**Published:** 2023-08-14

**Authors:** Wei Wu, Jia-Ning Lei, Qianzhuo Mao, Yan-Zhen Tian, Hong-Wei Shan, Jian-Ping Chen

**Affiliations:** State Key Laboratory for Managing Biotic and Chemical Threats to the Quality and Safety of Agro-Products, Key Laboratory of Biotechnology in Plant Protection of Ministry of Agriculture and Zhejiang Province, Institute of Plant Virology, Ningbo University, Ningbo 315211, China

**Keywords:** leafhopper, obligate symbionts, *Sulcia muelleri*, *Nasuia deltocephalinicola*, *Maiestas dorsalis*

## Abstract

**Simple Summary:**

Leafhoppers depend on plant sap as their food source, which is inherently unbalanced in terms of nutrition. To compensate for this deficiency, leafhoppers rely on obligate symbiotic bacterial associations to acquire the amino acids that are lacking in their diet. In this study, we focused on *Maiestas dorsalis* to understand the distribution of two obligate symbiotic bacteria, *Sulcia muelleri* and *Nasuia deltocephalinicola*, within the insect and their vertical transmission pathways. Our findings revealed their spatial arrangement within specialized tissues and provide insights into their genomic characteristics. We discovered that these bacteria undergo significant genome reduction but still retain the ability to synthesize essential amino acids for the leafhopper. This study enhances our understanding of the coevolutionary processes and nutritional interactions in Auchenorrhyncha insects, contributing to our knowledge of the intricate symbiotic relationships in nature.

**Abstract:**

Many insects rely on ancient symbiotic bacterial associations for essential nutrition. Auchenorrhyncha commonly harbor two obligate symbionts: *Sulcia* (Bacteroidetes) and a proteobacterial partner that supplies essential amino acids lacking in their plant-sap diets. In this study focusing on *Maiestas dorsalis*, we investigated the distribution and vertical transmission of two obligate symbiotic bacteria, *Sulcia* and *Nasuia*, within the leafhopper. *Sulcia* primarily inhabits the external region of the bacteriome, while *Nasuia* is restricted to the internal region. Both symbionts progressively infiltrate the ovary through the epithelial plug, ultimately reaching the developing primary oocyte. Furthermore, co-phylogenetic analysis suggests a close correlation between the evolution of Auchenorrhyncha insects and the presence of their obligate symbiotic bacteria. Genomic analysis further unveiled the extreme genome reduction of the obligate symbiotic bacteria, with *Sulcia* retaining genes involved in basic cellular processes and limited energy synthesis, while *Nasuia* exhibited further gene loss in replication, transcription, translation, and energy synthesis. However, both symbionts retained the genes for synthesizing the essential amino acids required by the host insect. Our study highlights the coevolutionary dynamics between *Sulcia*, proteobacterial partners, and their insect hosts, shedding light on the intricate nutritional interactions and evolutionary adaptations in Auchenorrhyncha insects.

## 1. Introduction

Microbial symbionts are ubiquitous in insects and play crucial roles in various aspects of insect biology including growth, development, reproduction, stress resistance, and adaptability [1,2,3,4]. Over long periods of evolution, some insects have developed stable symbiotic relationships with specific microbes known as obligate symbiotic microorganisms. These symbiotic partnerships help insects address nutrient deficiencies in their diets [1,2,5,6]. Among them, herbivorous insects obtain essential amino acids that are lacking in their normal diet through these symbiotic partners [1,2,7,8].

In the order Hemiptera, which includes insects with piercing-sucking mouthparts, plant sap serves as their main food source. However, plant sap is nutritionally imbalanced, being rich in sugars, inorganic compounds, and minerals, but deficient in amino acids [5,9]. Hemipteran insects often face inadequacies in essential amino acids, as plant sap predominantly contains non-essential amino acids such as glutamine, asparagine, and aspartic acid [7,9,10]. To overcome this dietary challenge, hemipteran insects commonly harbor one or more obligate symbiotic bacteria such as *Buchnera aphidicola* in aphids, *Carsonella ruddii* in psyllids, and *Portiera aleyrodidarum* in whiteflies [11,12,13]. These obligate symbiotic bacteria synthesize essential amino acids for the host insect’s survival by utilizing sugars and non-essential amino acids as substrates [5,9,14,15].

The suborder Auchenorrhyncha includes important agricultural pests such as leafhoppers, cicadas, spittlebugs, planthoppers, and treehoppers. Auchenorrhyncha usually harbor two types of obligate symbiotic bacteria: “*Candidatus Sulcia muelleri*” (referred to as *Sulcia*) and another symbiont from the phylum Proteobacteria such as “*Candidatus Baumannia cicadellinicola*” (*Baumannia*), “*Candidatus Zinderia* insecticola” (*Zinderia*), “*Candidatus* Nasuia deltocephalinicola” (*Nasuia*), or “*Candidatus* Hodgkinia cicadicola” (*Hodgkinia*) [16,17,18,19]. The association of Auchenorrhyncha with *Sulcia* and betaproteobacterial symbionts dates back more than 260 million years [17,20]. However, during the evolution of certain lineages of these insects, the beta-proteobacterium have been replaced by other bacteria. For instance, in some Cicadellinae species, the gammaproteobacterium *Baumannia* has taken over as the symbiont, while in certain species of Cicadoidea, the alphaproteobacteria *Hodgkinia* has taken over as the new symbiont [21,22,23,24,25]. In Auchenorrhyncha, *Sulcia* and its proteobacterial partners collaborate to provide the host with ten essential amino acids that are absent from their diet and cannot be synthesized by the insects themselves. Generally, *Sulcia*, as the primary symbiont, is responsible for synthesizing eight amino acids, while the secondary symbiont is responsible for synthesizing the remaining two essential amino acids [9,14,16,17].

*Maiestas dorsalis* (Hemiptera, Auchenorrhyncha, Cicadellidea) is an important rice pest and is found extensively in rice-producing regions across Asia. It poses a threat to rice plants by causing direct damage through feeding and serves as a vector insect for transmitting several rice virus pathogens such as rice gall dwarf virus, rice stripe mosaic virus, and rice tungro virus [26,27]. In our previous study, we analyzed the bacterial community structure and dynamics of *M. dorsalis* at different developmental stages using high-throughput sequencing [28]. We discovered that *Sulcia*-Md and *Nasuia*-Md serve as obligate symbiotic bacteria in *M. dorsalis* and are transmitted from the mother to the offspring via the ovary. In this study, we investigated the distribution and transovarial transmission of *Sulcia*-Md and *Nasuia*-Md in *M. dorsalis*. Additionally, we conducted a genomic analysis of *Sulcia* and *Nasuia* using high-throughput sequencing to examine their cooperative mechanism in providing essential amino acids to the host leafhopper.

## 2. Materials and Methods

### 2.1. Insect Rearing

Adult leafhoppers *M. dorsalis* were collected from a rice field in Jiaxing, Zhejiang Province, China, in September 2020. The leafhoppers were then reared in an insect-proof greenhouse for over two years under controlled conditions: 26 ± 1 °C temperature, a 16:8 h light-to-dark cycle, and 50 ± 5% relative humidity. TaiChung Native 1 (TN1) rice was cultivated in the same conditions for feeding the leafhoppers.

### 2.2. Tissue Sample Collection and Sequencing

To obtain bacteriocyte tissues for genomic DNA (gDNA) sequencing and investigate the genomic information of the two primary symbiotic bacteria, *Sulcia* and *Nasuia*. Approximately 500 adult leafhoppers of *M. dorsalis* were collected, and their bacteriomes were dissected in 95% ethanol. Total gDNA was extracted using the Qiagen DNeasy Extraction Kit according to the manufacturer’s instructions. For Illumina sequencing, three genomic DNA libraries were prepared with 500-bp, 5-kb, and 10-kb insertions using the Illumina Nextera XT DNA Sample Prep Kit. The gDNA samples were then sequenced on the Illumina MiSeq platform using the MiSeq Reagent Kit v3 by Novogene Corporation Inc (Novogene, Beijing, China).

### 2.3. Genome Assembly and Annotation

Quality checks of the raw reads were performed using FastQC [29], followed by adapter trimming with Trimmomatic [30]. The genome was de novo assembled with Velvet 1.2.10 [31], and putative contigs were identified by tblastx alignments to proteins from the published *Sulcia* and *Nasuia* genomes. To fill gaps, we utilized paired-end long insert reads and mate-pair reads with SSPACE [32]. Further enhancement of the genome assembly was conducted using Pilon [33]. Genome completeness was evaluated with BUSCO v2, which utilizes predefined lineage-specific sets of benchmarking universal single-copy orthologs (BUSCOs) as a reference for gene content expectations. However, considering the well-documented natural process of genome reduction in both *Sulcia* and *Nasuia* [17], we included four representative species of *Sulcia* and two representative species of *Nasuia* for comparison (Appendix A). Gene prediction for the assembled genomes of *Sulcia* and *Nasuia* utilized prodigal, RNAmmer, RNAscan-SE, Rfam, and RepeatMasker software. The predicted amino acid sequences were compared to various databases (COG, GO, KEGG, NR, and Swiss-Prot) using NCBI BLAST. Annotations of genes and their functions were combined to generate the annotation for *Sulcia* and *Nasuia*. Circos software was used to display the genome, analyze noncoding RNA, and provide gene function annotations, constructing a genome-wide map of the strain.

### 2.4. Phylogenetic Analysis

To investigate the phylogenetic relationship between *Sulcia* and its proteobacterial partner and other Auchenorrhyncha, maximum likelihood (ML) and Bayesian inference (BI) tree reconstructions were performed. A total of 52 16S rRNA sequences of *Sulcia* and *Nasuia* from other Auchenorrhyncha insects were downloaded from the NCBI database. The 16S gene sequences were aligned using MAFFT with default parameters [34]. ModelFinder was used to determine the best-fit model using the Bayesian Information Criterion (BIC). The GTR + F + I + G4 model was chosen for Bayesian analysis, while the GTR + I + G model was selected for IQ-TREE. In the BI analysis, the settings included 2,000,000 Markov chain Monte Carlo (MCMC) generations, sampling frequency of 100, and a burn-in of 25%. Maximum likelihood (ML) analyses were conducted using a heuristic search method (10,000 random addition replicates tree-bisection-reconnection, TBR, branch swapping) with 1000 bootstrap replications. PhyloSuite was used for MAFFT, ModelFinder, IQ-TREE, Ultrafast bootstrap, and MrBayes [35,36,37,38].

### 2.5. Fluorescence In Situ Hybridization (FISH)

To investigate the distribution of *Sulcia* and *Nasuia* within the body of *M. dorsalis*, 30 ovaries and bacteriomes from adult female *M. dorsalis* at different days post-emergence were dissected. The dissected samples were fixed overnight in 4 °C paraformaldehyde. After fixation, the samples underwent pre-treatment in hybridization buffer (20 mM Tris-HCl, 180 mM NaCl, 10% *v*/*v* SDS, 30% *v*/*v* formamide) for 15 min. Subsequently, the samples were incubated in hybridization buffer containing 10 nM oligonucleotide DNA probes targeting the 16S rRNA sequences of *Sulcia*-cy3 (5′-CTG AAT TAC AACGTA CAA AAC CC-3′-Cy3) and *Nasuia*-FITC (5′-GTA CTA ATT CTT TTA CAA GCA CTT-3′-FITC) (Sanggong, Shanghai, China) as previously mentioned, with minor adjustments. The incubation took place at 50 °C for 4 h, followed by thorough washing with wash buffer (0.15 M NaCl, 0.015 M sodium citrate). The samples were examined using a Leica TCS SP8 confocal microscope (Leica Microsystems, Wetzlar, Germany).

### 2.6. Transmission Electron Microscopy

To investigate the subcellular distribution of *Sulcia* and *Nasuia* within the ovaries and bacteriomes of *M. dorsalis*, 40 ovaries and bacteriomes from adult female *M. dorsalis* at different days post-emergence were dissected. The samples were initially fixed with a solution of 2% paraformaldehyde and 2.5% glutaraldehyde in PBS buffer at 4 °C overnight. After several washes with PBS buffer, they were further fixed with 2% osmium tetroxide in PBS buffer at 4 °C overnight. Following additional washes in PBS buffer, the fixed tissues underwent dehydration using a series of ethanol concentrations (30%, 50%, 70%, 80%, 90%, 95%, and 100%) for 20 min each. The samples were then transferred to absolute acetone for an additional 20 min. Subsequently, the samples were immersed in a 1:1 mixture of Spurr resin and absolute acetone at room temperature for 1 h, followed by a transfer to a 3:1 mixture of Spurr resin and absolute acetone at room temperature for 3 h. Finally, the samples were placed with absolute Spurr resin overnight. The prepared samples were embedded in capsules containing an embedding medium and heated at 70 °C overnight. For visualization of the specimen sections, they were stained with uranyl acetate and alkaline lead citrate for 5–10 min each. The resulting images were obtained using a Hitachi electron microscope HT7800.

## 3. Results

Distribution of *Sulcia*-Md and *Nasuia*-Md in the bacteriome of the leafhopper *M. dorsalis*:

Microscopic examination revealed the presence of a pair of oval or kidney-shaped bacteriomes on both sides of the anterior abdomen segments 1–3 of *M. dorsalis*, ranging from approximately 0.2 to 0.4 mm in length. The bacteriomes had a white, opaque appearance (Figure 1A,B). The FISH results demonstrated distinct spatial distributions of the co-obligate symbiotic bacteria within the bacteriomes. *Sulcia*-Md was predominantly localized in the outer region of the bacteriome, while *Nasuia*-Md was restricted to the inner region (Figure 1C–E). This distribution pattern was further confirmed through three-dimensional reconstruction using confocal microscopy, where *Nasuia*-Md occupied the central region of the bacteriomes and was enveloped by *Sulcia*-Md in the outer region (Figure 1F and Appendix A). Electron microscopy observations also supported these findings, showing a predominant distribution of darker-colored *Sulcia*-Md in the periphery of the lighter-colored *Nasuia*-Md within the bacteriome.

### 3.1. Vertical Transmission Pathway of Sulcia*-Md* and Nasuia*-Md* in M. dorsalis

The female reproductive system of *M. dorsalis* consists of a pair of ovaries, each containing multiple ovarian tubules. These tubules can be distinguished from top to bottom as the terminal filament, germarium, ovarian tubules, and pedicel. The ovarian tubules contain oocytes at different stages of development and are surrounded by a layer of follicle cells. Between the primary oocytes and the pedicel, there is a specialized type of follicle cells known as the epithelial plug (Figure 2A). The FISH results revealed a gradual invasion of *Nasuia*-Md and *Sulcia*-Md into the ovaries of adult female *M. dorsalis*. At the early stages of ovarian development, no discernible distribution of the symbiotic bacteria was observed. As the ovary matured, both *Sulcia*-Md and *Nasuia*-Md progressively infiltrated the ovary through the epithelial plug, ultimately reaching the developing primary oocyte located at the base of the ovarian tube (Figure 2B,D). Finally, a “symbiont ball” was formed at the posterior end of the primary oocyte (Figure 2C,E). Electron microscopy also confirmed the entry of *Sulcia*-Md and *Nasuia*-Md into the ovary through the epithelial plug, ultimately forming a “symbiont ball” at the posterior pole of the terminal oocyte (Figure 2F–I).

### 3.2. Co-Phylogenetic Analysis of Sulcia and Its Proteobacterial Partners

To investigate the phylogenetic relationship between *Sulcia* and its proteobacterial partners in *M. dorsalis* and other Auchenorrhyncha insects, we conducted maximum likelihood (ML) and Bayesian inference (BI) tree reconstructions. We obtained 16S rRNA sequences of *Sulcia* and its proteobacterial partners (including *Nasuia*, *Baumannia*, *Zinderia*, and *Hodgkinia*) from 52 insects belonging to three superfamilies, seven families, and 45 genera within Auchenorrhyncha (Appendix A). Phylogenetic analysis of *Sulcia* 16S rRNA revealed a consistent evolutionary relationship between *Sulcia* and Auchenorrhyncha, supporting previous findings. Within the superfamily Membracoidea, two distinct branches were identified, primarily composed of insects from the subfamilies Cicadellinae and Deltocephalinae. Notably, *Baumannia* and *Nasuia* were found to be the respective proteobacterial partners associated with these subfamilies (Figure 3, Appendix A). The evolution of *Sulcia* and its proteobacterial partners generally shows consistency, but there are differences observed among insect hosts of different species within the same subfamily (Figure 3). This suggests a close correlation between the evolution of Auchenorrhyncha insects and the presence of their obligate symbiotic bacteria. The mutual replacement of proteobacterial partners may be indicative of variations in the host’s food sources and nutritional requirements.

### 3.3. Co-Obligate Symbiont Genome Characteristics for M. dorsalis

The genome sizes of *Sulcia*-Md and *Nasuia*-Md were determined to be 205,313 and 121,476 bp, respectively (Figure 4 and Table 1). Both symbionts exhibited low GC content, which is typical in obligate symbioses: *Sulcia*-Md had a GC content of 24%, while *Nasuia*-Md had a GC content of 16% (Table 1). *Sulcia*-Md contained 215 predicted protein-coding sequences (CDS) including 21 encoding hypothetical proteins. It also possessed three ribosomal cassettes, 31 tRNAs, and 10 CDS that were truncated or pseudogenized with uncertain function (Figure 4A and Table 1). *Nasuia*-Md, on the other hand, had 168 CDS, with 144 encoding hypothetical proteins. It had two ribosomal cassettes, 30 tRNAs, and one identifiable pseudogene (Figure 4B and Table 1). Our assembly was complete as the proportion of missing BUSCO marker genes in the *Sulcia*-Md draft assembly and *Nasuia*-Md draft assembly fell well within the range of the previously completely sequenced *Sulcia* and *Nasuia* genomes, respectively (Appendix A).

The examination of the gene content in *Sulcia*-Md and *Nasuia*-Md revealed a substantial level of gene loss in both bacteria. In *Sulcia*-Md, only genes associated with fundamental processes such as replication, transcription, and translation were retained. These included genes encoding subunits of DNA and RNA polymerases, enzymes involved in RNA modification, ribosomal proteins, and a limited number of genes involved in energy synthesis (Figure 5). In contrast, *Nasuia*-Md exhibited an even greater degree of gene loss. It not only lacked a significant number of genes related to replication, transcription, and translation, but also all genes associated with energy synthesis (Figure 5).

### 3.4. Analysis of the EEA Synthesis Pathway of the Obligate Symbiotic Bacteria Sulcia*-Md* and Nasuia*-Md* in M. dorsalis

Previous investigations have demonstrated that the obligate symbiotic bacteria *Sulcia*-Md and its proteobacterial partners in Auchenorrhyncha insects primarily play a role in providing the host insects with ten essential amino acids (EAA) [17]. Therefore, we conducted an analysis of the amino acid synthesis pathways in *Sulcia*-Md and *Nasuia*-Md of *M. dorsalis*. *Sulcia*-Md retained the synthesis pathways for eight essential amino acids: threonine, isoleucine, lysine, leucine, valine, arginine, phenylalanine, and tryptophan. However, it lacked two proteins involved in the lysine synthesis pathway, namely, succinyl-diaminopimelate desuccinylase (DapE) and diaminopimelate decarboxylase (LysA). Additionally, it exhibited a deficiency in the enzyme argininosuccinate lyase (ArgH), which is essential for the final step of the arginine synthesis pathway.

*Nasuia*-Md possessed the ability to synthesize two EAA, histidine, and methionine. The synthesis of methionine involved cooperative efforts between *Nasuia*-Md and *Sulcia*-Md, with *Nasuia*-Md utilizing homoserine, an intermediate product of the threonine synthesis pathway in *Sulcia*-Md, as a precursor for methionine synthesis. However, both the histidine and methionine synthesis pathways in *Nasuia*-Md were incomplete. The histidine synthesis pathway lacked two vital proteins, phosphoribosyl-AMP cyclohydrolase (HisL) and histidinol-phosphate aminotransferase (HisC). Similarly, the methionine synthesis pathway was deficient in homoserine O-succinyltransferase (MetA) and cysteine-S-conjugate beta-lyase (MetC).

## 4. Discussion

Auchenorrhyncha insects have adapted to a diet of nutrient-deficient plant sap. Through a lengthy evolutionary process, they have established complex symbiotic relationships with at least two obligate bacterial symbionts to obtain essential amino acids that are lacking in their diet [9,14,16,17]. Approximately 260 to 280 million years ago, the ancestors of Auchenorrhyncha formed a symbiotic relationship with a *Sulcia* and betaproteobacterial symbionts [17,20]. As this group diversified, many lineages acquired additional symbionts from the proteobacterial phylum. *Sulcia* and its proteobacterial partners reside in specialized insect-specific tissues known as bacteriomes and are vertically transmitted through the ovaries to ensure their transmission across generations [17,20,39]. This study focused on the symbiotic associations in *M. dorsalis*. Our findings revealed that *Sulcia*-Md primarily inhabits the outer region of the bacteriomes, while *Nasuia*-Md is confined to the inner region (Figure 1). Both *Sulcia*-Md and *Nasuia*-Md follow a previously reported pathway to enter the ovaries through the epithelial plug and eventually reach the developing mature oocytes (Figure 2). Interestingly, the bacteriomes of *M. dorsalis* exhibited a milky-white color (Figure 1B), which was in contrast to the pale yellow color observed in the bacteriomes of *Nephotettix cincticeps* and *Dalbulus maidis* [39,40].

The primary endosymbiont in Auchenorrhyncha, *Sulcia*, is believed to have originated from a single ancient infection event [17,20]. Recent studies have demonstrated a strong correlation between the phylogenies of *Sulcia* and its host insects, indicating a co-diversification process that has occurred over millions of years [41]. In ancestral Auchenorrhynchans, *Sulcia* co-resided with another symbiont from the class Betaproteobacteria [17,20]. Among the many phloem-feeding auchenorrhynchans, the presence of ancestral Betaproteobacterial symbionts is prevalent. For instance, in most leafhoppers, *Sulcia* co-occurs with *Nasuia*; in most froghoppers with *Zinderia*; in some planthoppers with *Candidatus* Vidania fulgoroidea [10,17,18,19,39]. However, the scenario changes for xylem-feeding auchenorrhynchans, where a gradual transition to different proteobacterial partners is observed. Cicadas co-occur with *Hodgkinia* (Alphaproteobacteria), sharpshooters with *Baumannia* (Gammaproteobacteria), and spittlebugs with *Sodalis*-like symbiont (Gammaproteobacteria) [42,43,44]. Notably, certain auchenorrhynchans have a third associate in addition to the bacterium *Sulcia* and its co-symbiont. In leafhopper *Ledropsis discolor*, *Sulcia* is accompanied by yeast-like symbionts. Likewise, in some leafhopper species and Delphacidae planthoppers, ancestral bacterial symbionts have been replaced by yeast-like symbionts [25,45,46]. This suggests that the replacement of symbionts was continuous and independent during the evolution of Auchenorrhynchans. As a result of this dynamic symbiosis, the phylogenies of *Sulcia* and its proteobacterial partners show rough correlations. However, there is also some variation in the phylogeny within the group, likely due to the changing associations between the auchenorrhynchans and their symbionts over evolutionary time. This complexity underscores the intricate relationships between Auchenorrhynchans and their symbiotic partners, which have shaped their evolutionary history in unique ways (Figure 3, Appendix A).

In obligate bacterial symbionts, a close and specialized association with their insect hosts is observed as they reside within specific cells or tissues. These symbionts rely on crucial cellular functions of the host for their survival and are transmitted vertically across insect host generations to maintain their presence [1,2,8]. One notable feature of this symbiotic relationship is the significant genome reduction observed in obligate symbiotic bacteria. This reduction is a consequence of their long-term residence within specialized host cells (bacteriomes) and their vertical transmission over millions of years [8,9,10,17]. In some extreme cases, obligate symbionts retain only a minimal genetic capacity, sufficient for essential life processes such as replication, transcription, and translation [17,47,48,49]. Despite this genomic minimalism, they manage to retain genes associated with the synthesis pathways of essential nutrients required by the host [17,47,48]. The genomic analysis of *Sulcia*-Md and *Nasuia*-Md supports these characteristics. The *Sulcia*-Md genome retains genes related to fundamental processes such as replication, transcription, and translation, along with a limited number of genes involved in energy synthesis (Figure 5). In contrast, *Nasuia*-Md lacks a significant number of genes related to replication, transcription, and translation as well as all genes associated with energy synthesis (Figure 5). However, both *Sulcia*-Md and *Nasuia*-Md genomes maintain the majority of genes involved in the synthesis pathways of the 10 essential amino acids required by the insect host (Figure 6). In summary, the close association and genome reduction observed in obligate bacterial symbionts highlight the coevolutionary dynamics between these symbionts and their insect hosts, ultimately leading to their interdependence and the retention of essential functional traits.

In Auchenorrhyncha insects, *Sulcia* and its proteobacterial partners collaborate to provide the insect host with 10 essential amino acids [42,50]. Typically, *Sulcia* is responsible for synthesizing eight essential amino acids (leucine, isoleucine, threonine, lysine, arginine, tryptophan, phenylalanine, and valine), while the proteobacterial partners (such as *Nasuia*, *Baumannia*, and *Hodgkinia*) are responsible for synthesizing the remaining two essential amino acids (histidine and methionine) and riboflavin (vitamin B2) [17,50,51,52]. In the spittlebug, *Zinderia* can synthesize three essential amino acids (methionine, histidine, and tryptophan) and riboflavin (vitamin B2), while *Sulcia* synthesizes the remaining seven essential amino acids [53]. In the planthoppers *Purcelliella pentastirinorum* and *Oliarus filicicola*, *Sulcia* is only responsible for synthesizing leucine, isoleucine, and valine, while the remaining seven amino acids are synthesized by *Vidania fulgoroidea* [51,52]. These variations in amino acid synthesis combinations may result from adaptive changes that have occurred during the coevolution of different host insects and their symbiotic bacteria to meet the specific nutritional requirements of the host.

In *M. dorsalis*, the obligate symbiont *Sulcia*-Md retains most of the genes necessary for synthesizing eight EAAs. However, it lacks some key genes, specifically DapE and LysA in the lysine synthesis pathway as well as ArgH in the arginine synthesis pathway. Similarly, *Nasuia*-Md is responsible for synthesizing two essential amino acids, histidine and methionine. Nevertheless, both the histidine and methionine synthesis pathways are incomplete due to the absence of crucial proteins HisL and HisC in the histidine pathway as well as MetA and MetC in the methionine pathway. It is not uncommon for insect symbiotic genomes to lose essential genes involved in critical initiation, intermediate, and final catabolic steps in nutritional synthesis [54]. Recent research has provided insights into a possible mechanism that compensates for the missing genes in the essential amino acid synthesis pathways of symbiotic bacteria. Studies on the primary symbiont *Buchnera* of aphids have revealed that it lacks a critical gene, aspartate aminotransferase (aspC), in the phenylalanine synthesis pathway [7,54]. Interestingly, transcriptomic data from the pea aphid-*Buchnera* symbiosis indicate that insect-encoded genes for aspartate aminotransferase are upregulated in the bacterial symbiont [7,54]. This upregulation suggests that the insect-encoded proteins may complement the incomplete phenylalanine biosynthetic pathway in *Buchnera*. Moreover, in other leafhoppers belonging to the Cicadellidae family, *Nasuia* is known to contribute to the synthesis of riboflavin (vitamin B2). However, our analysis did not reveal the presence of the riboflavin synthesis pathway in *Nasuia*-Md of *M. dorsalis*. This discrepancy in riboflavin synthesis could potentially explain the variation in color between the bacteriome of *M. dorsalis* and other leafhoppers, as riboflavin is recognized for its orange-yellow pigmentation [14]. In summary, this study provides a detailed description of the distribution of two obligate symbiotic bacteria, *Sulcia*-Md and *Nasuia*-Md, within the leafhopper *M. dorsalis* as well as their vertical transmission process within the insect. Additionally, the study investigated their nutritional symbiotic relationship with the host insect. These findings contribute to a deeper understanding of the coevolutionary processes and nutritional interactions in Auchenorrhyncha, contributing to our knowledge of the intricate symbiotic relationships in nature.

## Figures and Tables

**Figure 1 insects-14-00710-f001:**
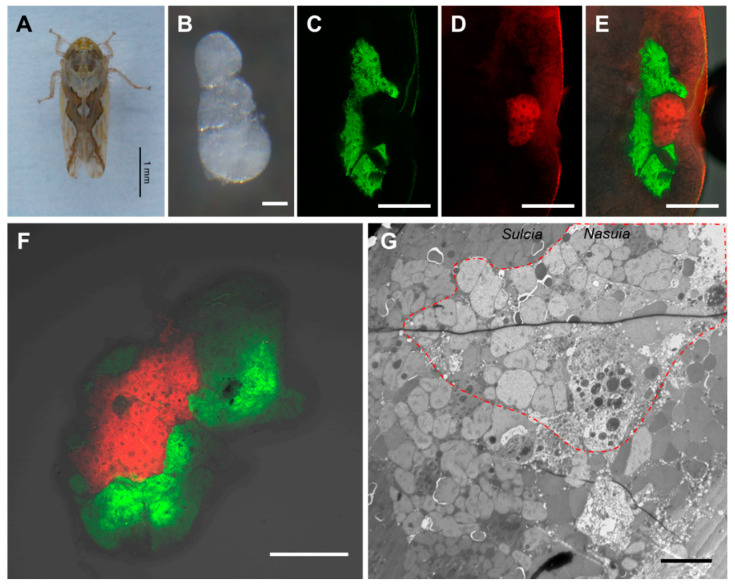
The distribution of *Sulcia* and *Nasuia* in the bacteriome of the leafhopper *M. dorsalis.* (**A**) Adult male leafhopper *M. dorsalis*. Bar, 1 mm. (**B**) The dissected bacteriome of *M. dorsalis*. Bar, 100 µm. (**C**–**E**) The distribution of *Sulcia* (green) and *Nasuia* (red) in the bacteriome of *M. dorsalis* detected by FISH. Bar, 100 µm. (**F**) The distribution of *Sulcia* and *Nasuia* in the dissected bacteriome of *M. dorsalis* detected by FISH. Bar, 100 µm. (**G**) The distribution of *Sulcia* (the region outside the red dashed lines) and *Nasuia* (the region within the red dashed lines) in the bacteriome of *M. dorsalis* was observed using TEM. Bar, 10 µm. All images are representative of at least three replicates.

**Figure 2 insects-14-00710-f002:**
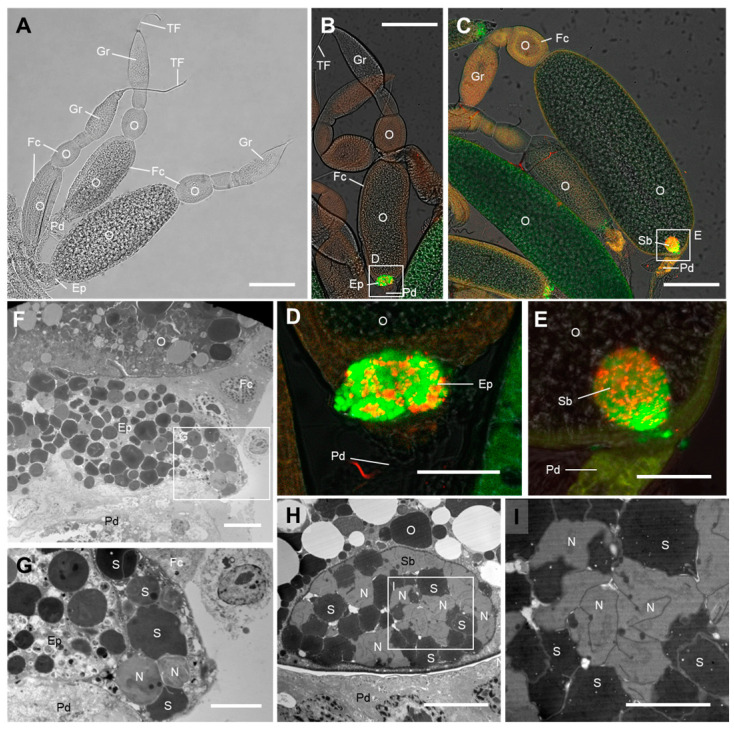
The process of *Sulcia* and *Nasuia* entering the oocyte in adult female *M. dorsalis.* (**A**) The ovarian structure of the leafhopper *M. dorsalis*. Bar, 200 μm. (**B**–**E**) The distribution of *Sulcia* and *Nasuia* within the ovarian epithelial plug as well as their entry into the posterior pole of mature oocytes in female ovaries, resulting in the formation of a “symbiont ball”. Scale bars in (**B**,**C**): 200 µm; (**D**,**E**): 50 µm. (**D**,**E**) are enlargements of the boxed area in (**B**,**C**), respectively. (**F**–**I**) TEM observations revealed the distribution of *Sulcia* and *Nasuia* within the ovarian epithelial plug and the symbiont ball inside the primary oocytes. Scale bars in (**F**,**H**): 10 µm; (**G**,**I**): 5 µm. I is an enlargement of the boxed area in H. TF, terminal filament; Gr, germarium; O, oocyte; Fc, follicular cell; Ep, epithelial plug; Pd, pedicel; Sb, symbiont ball; S, *Sulcia*; N, *Nasuia*. All images are representative of at least three replicates.

**Figure 3 insects-14-00710-f003:**
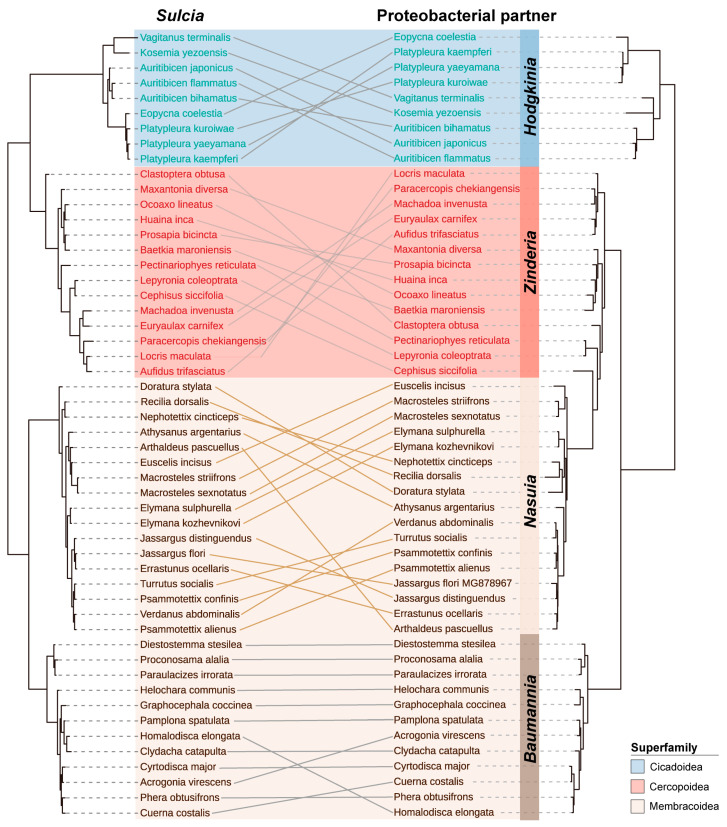
Co-phylogenetic analysis of “Ca. *Sulcia muelleri*” (**left**) and its proteobacterial partners (**right**) based on the Bayesian inference tree. Terminals of *Sulcia* and its proteobacterial partner were labeled with the corresponding host taxon name. The proteobacterial partners were additionally labeled with the specific bacterial name on the right.

**Figure 4 insects-14-00710-f004:**
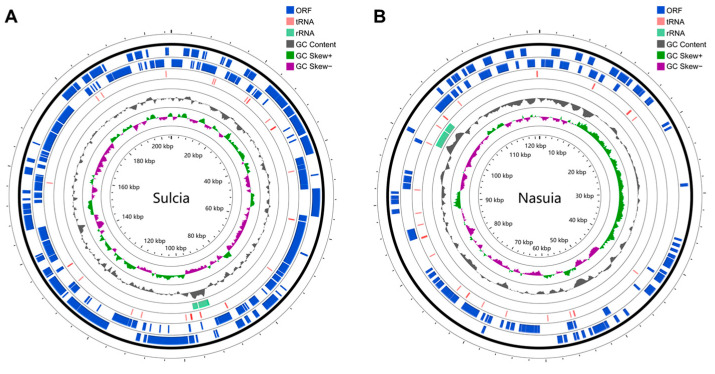
Circular representation of the genomes of co-obligate symbiotic bacteria, *Sulcia* (**A**) and *Nasuia* (**B**), in *M. dorsalis*. In the chromosomal DNA map, from the outermost to inner, the circles show (1) protein-coding genes on the forward strand and reverse strand; (2) tRNA genes; (3) rRNA genes; (4) GC content; (5) GC skew; (6) scale marks of the genome.

**Figure 5 insects-14-00710-f005:**
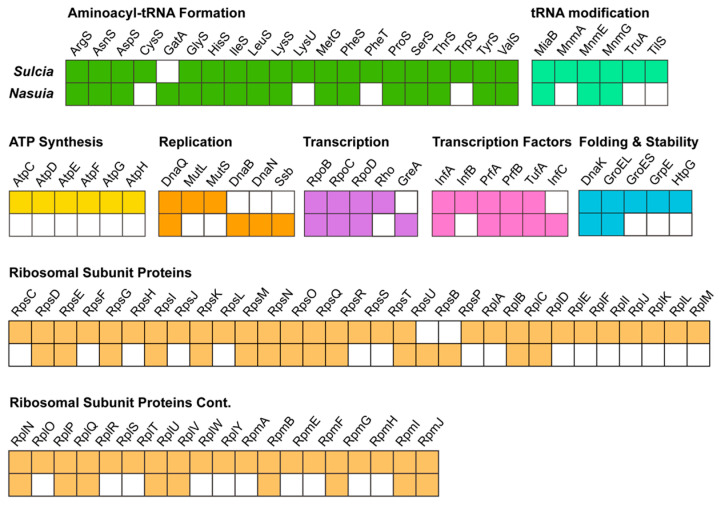
Genomic content of the co-obligate symbiotic bacteria, *Sulcia* and *Nasuia*, in *M. dorsalis*. The major pathways include DNA replication, translation and processing, protein stability, and energy metabolism. Colored boxes indicate genes that are present, and white boxes indicate gene absence.

**Figure 6 insects-14-00710-f006:**
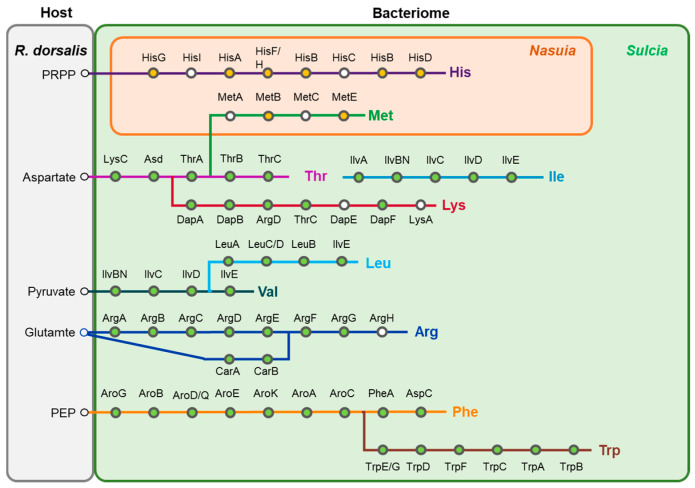
The biosynthetic pathways of the 10 EAAs in the co-obligate symbiotic bacteria, *Sulcia* and *Nasuia*, in *M. dorsalis*. Each circle represents a step in the pathway, with gene abbreviations displayed on top. Colored circles indicate the presence of the corresponding gene, while white circles represent the absence of the gene. Green circles and yellow circles represent genes encoded by *Sulcia* and *Nasuia*, respectively.

**Table 1 insects-14-00710-t001:** Genome statistics of *Sulcia*-Md and *Nasuia*-Md.

Feature	*Sulcia*-Md	*Nasuia*-Md
Genome size (bp)	205,313 bp	121,476 bp
G + C content (%)	24	16
Protein-coding genes (CDS)	215	168
CDS average length (bp)	842	367
Percent of coding region (%)	88.18	50.8
rRNA (5S, 16S, 23S)	3	2
tRNA	31	30
Genes with function prediction	194	24

## Data Availability

Data are deposited at the China National Microbiology Data Center (NMDC) with accession numbers NMDC60105950 (https://nmdc.cn/resource/genomics/project/detail/NMDC60105950, accessed on 6 August 2023) and NMDC60105951 (https://nmdc.cn/resource/genomics/project/detail/NMDC60105951, accessed on 6 August 2023).

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
