# Peer review of "Distribution, Vertical Transmission, and Cooperative Mechanisms of Obligate Symbiotic Bacteria in the Leafhopper Maiestas dorsalis (Hemiptera, Cicadellidea)"

_insects, 2023, doi:10.3390/insects14080710_

Round 1

Reviewer 1 Report

Please see attached file. Lot of work has gone into this study but the discussion does not highlight any novel findings, and it seems to be a very confirmatory study following on from the authors more interesting previous work.

The discussion is just another introduction and literature review. Please try and improve this especially.

I have corrected the English language in general but maybe your ms should be re-checked.

Author Response

We extend our gratitude for the reviewer's insightful comments. In response to their feedback, we have diligently rectified the grammar and writing errors present in the manuscript. Moreover, we have addressed any ambiguities found in the Materials and Methods section. Additionally, we have undertaken substantial revisions in the Discussion section, carefully considering the reviewer's recommendations. 

Specific details can be found in the revised manuscript and the point-by-point response to the review. We have thoroughly addressed each of the reviewer's comments and made the necessary revisions accordingly. The revised manuscript contains all the changes made to address the concerns raised by the reviewer. Additionally, the point-by-point response to the review document provides a detailed explanation of how we have addressed each specific comment.

We are confident that the revisions have significantly improved the quality and clarity of the manuscript. We are grateful for the reviewer's constructive feedback, which has contributed to enhancing the overall value of our research.

Reviewer 2 Report

In this study, you show the localization of Sulcia and Nasuia in the bacteriome of Recilia dorsalis, the co-phylogenetic history of endosymbionts infecting Auchenorrhyncha insects, and with de novo genome assemblies reveal the contributions of each symbiont to the production of essential amino acids for the host. I have only minor comments with respect to the Methods/Results (including some typos) and several questions that should be addressed in the Discussion.

L108-L118: Some of the comments here apply to the corresponding results section. How were symbiont reads (and/or contigs) distinguished from host reads (and/or contigs)? Were the genomes circularized using a specific locus, or were they recovered linearly and just presented as circular?

L210-L212: Yes, broadly correlated phylogenies of co-symbiotic partners, but there is also substantial discordance of phylogenies within groups (suggestive of host-switching). This seems worth mentioning briefly here in the results or in the discussion.

L215-L223: There are several important considerations that were not mentioned here. First, were the genomes recovered in a single contig (this should be a row in table 1) that was circularized? What was the BUSCO score for these genomes, and what database was used in running BUSCO (this is particularly important for the inferences regarding the amino acid synthesis pathways)?

L248-L253: This observation should be discussed in the Discussion. How is Nasuia able to produce these essential amino acids without the full pathways? This is also why a reported BUSCO score is important. Could these loci have been left out of the assembly? Ditto for incomplete synthesis pathways for Arginine and Lysine in Sulcia.

L275 Figure 3: The legend reads “Supplefamily” where it should read “Superfamily”. The lines connecting Baumania and Sulcia co-symbionts look fragmented. This appears to be a formatting error. I am guessing these are all supposed to be horizontal lines (excepting the several diagonal connections).

Author Response

Review 2:

In this study, you show the localization of Sulcia and Nasuia in the bacteriome of Recilia dorsalis, the co-phylogenetic history of endosymbionts infecting Auchenorrhyncha insects, and with de novo genome assemblies reveal the contributions of each symbiont to the production of essential amino acids for the host. I have only minor comments with respect to the Methods/Results (including some typos) and several questions that should be addressed in the Discussion.

Response: We greatly appreciate your interest and positive comments to our work. The manuscript has been carefully modified and improved following your suggestions. Please see our responses below.

L108-L118: Some of the comments here apply to the corresponding results section. How were symbiont reads (and/or contigs) distinguished from host reads (and/or contigs)? Were the genomes circularized using a specific locus, or were they recovered linearly and just presented as circular?

Response: Thank you for your feedback. We aligned the contigs obtained through Velvet assembly with the published genomes of Nasuia and Sulcia to retrieve the genomic sequences originating from Sulcia and Nasuia. To address gaps in the assembly, we utilized paired-end long insert reads and mate-pair reads with SSPACE. As a result, we obtained a linear representation of the genome, which we subsequently reconstructed into a circular form. For more comprehensive information, please refer to the revised manuscript, lines 101-105, and lines 108-111.

L210-L212: Yes, broadly correlated phylogenies of co-symbiotic partners, but there is also substantial discordance of phylogenies within groups (suggestive of host-switching). This seems worth mentioning briefly here in the results or in the discussion.

 Response: Thank you for your comments. Based on your feedback, we have made revisions to the Results and Discussion sections concerning the evolutionary relationship between Sulcia and its proteobacterial partner. For specific details, please refer to the revised manuscript, lines 215-220, and lines 348-368.

L215-L223: There are several important considerations that were not mentioned here. First, were the genomes recovered in a single contig (this should be a row in table 1) that was circularized? What was the BUSCO score for these genomes, and what database was used in running BUSCO (this is particularly important for the inferences regarding the amino acid synthesis pathways)?

 Response: Thank you for your feedback. We have taken your comments into serious consideration and have made the necessary changes to the manuscript. Specifically, we have provided a detailed description of our method for genome completeness assessment using BUSCO in the Materials and Methods section. Additionally, we have included a description of the BUSCO results in the Results section, along with the addition of Supplementary Table 1 and Supplementary Figure 3. These revisions have enhanced the clarity and completeness of our work. We hope that these modifications address your concerns, and we sincerely appreciate your valuable input. For specific details, please refer to the revised manuscript, lines 112-117 and lines 231-234, as well as Supplementary Table 1 and Supplementary Figure 3.

L248-L253: This observation should be discussed in the Discussion. How is Nasuia able to produce these essential amino acids without the full pathways? This is also why a reported BUSCO score is important. Could these loci have been left out of the assembly? Ditto for incomplete synthesis pathways for Arginine and Lysine in Sulcia.

 Response: Thank you for your valuable comments. We have carefully considered your feedback and made revisions to the Discussion section accordingly. Based on the existing literature, we now propose that the absence of Sulcia and Nasuia genes in the EEA synthesis pathway might have been compensated for by host-encoded proteins, which could include pre-existing eukaryotic homologs and/or genes of bacterial origin acquired through horizontal gene transfer. This hypothesis sheds light on potential mechanisms underlying the observed phenomenon. For specific details, please refer to the revised manuscript, specifically lines 405-421. We sincerely appreciate your constructive review and the opportunity to improve our research accordingly.

L275 Figure 3: The legend reads “Supplefamily” where it should read “Superfamily”. The lines connecting Baumania and Sulcia co-symbionts look fragmented. This appears to be a formatting error. I am guessing these are all supposed to be horizontal lines (excepting the several diagonal connections).

Response: We are grateful for the reviewer's reminder. We have addressed the issues in the figures and made the necessary corrections in the revised manuscript.

Reviewer 3 Report

The manuscript by Wu et al investigated the distribution of vertical transmission of two obligate symbiotic bacteria Sulcia and Nasuia in rice pest Recilia dorsalis. They did genomic and phylogenetic analysis and performed FISH to document and understand the underlying mechanisms. The study is systematically conducted with a good number of replicates. The manuscript is well written, and I had very few minor edits.  

 Minor edits: 

Italicizing Nausia-Rd, Sulcia and R. dorsalis at several places. The correction should be made in the sub-headings (lines: 166, 182, 243 etc) and figure legends (Eg: line 299) and elsewhere in the manuscript. 

Line 214 and 234: Please change Recilia dorsalis to R. dorsalis 

Line 233, 237, 245: Abbreviation of Essential Amino Acids “Is it EAA or EEA. Please correct. Fig 6 says EAA. 

Author Response

The manuscript by Wu et al investigated the distribution of vertical transmission of two obligate symbiotic bacteria Sulcia and Nasuia in rice pest Recilia dorsalis. They did genomic and phylogenetic analysis and performed FISH to document and understand the underlying mechanisms. The study is systematically conducted with a good number of replicates. The manuscript is well written, and I had very few minor edits.  

 Response: We greatly appreciate your interest and positive comments to our work. The manuscript has been carefully modified and improved following your suggestions. Please see our responses below.

Minor edits: 

Italicizing Nausia-Rd, Sulcia and R. dorsalis at several places. The correction should be made in the sub-headings (lines: 166, 182, 243 etc) and figure legends (Eg: line 299) and elsewhere in the manuscript. 

Line 214 and 234: Please change Recilia dorsalis to R. dorsalis.  

Response: We sincerely appreciate the reviewer's valuable input. In response to the feedback, we have rectified the writing errors in the revised manuscript. Furthermore, we conducted a meticulous review of the entire paper, ensuring a comprehensive correction of all identified writing errors.

Line 233, 237, 245: Abbreviation of Essential Amino Acids “Is it EAA or EEA. Please correct. Fig 6 says EAA. 

Response: We are grateful for the reviewer's valuable feedback. You are absolutely correct, and we apologize for the oversight in using the incorrect abbreviation for essential amino acids. In the revised manuscript, we have made the necessary modifications to rectify this error. Thank you for bringing this to our attention once again.

Reviewer 4 Report

An interesting MS on still insuffciently studied issue of endosymbiotic bacteria in insects. While I am not specialisit mcirobiologist and cannot evaluate methodology properly, I have few small issues concerning the MS:

1. why insects must have been dissected to do the study. In other hemipterans e.g. (Kaszyca-Taszakowska & Depa 2022) the genomic samples of bacterial symbionts where extracted directly from whole aphid specimens, whithout dissecting. Is there any possibility of contamination of the sample in this way? Please state it shortly in the revised version of the MS. 

2. at line 321 there is 2.8 billion yers which is absolutely impossible. Please correct it and provide correct age of this symbiosis. 

3. Is there any phylogenetic relation of Sulcia with other hemipterans e.g. Buchnera in aphids, or these are rather evolutionary independent and separate infestations of unrelated bacteria?

Author Response

An interesting MS on still insuffciently studied issue of endosymbiotic bacteria in insects. While I am not specialisit mcirobiologist and cannot evaluate methodology properly, I have few small issues concerning the MS:

 Response: We greatly appreciate your interest and positive comments to our work. The manuscript has been carefully modified and improved following your suggestions. Please see our responses below.

  1. why insects must have been dissected to do the study. In other hemipterans e.g. (Kaszyca-Taszakowska & Depa 2022) the genomic samples of bacterial symbionts where extracted directly from whole aphid specimens, whithout dissecting. Is there any possibility of contamination of the sample in this way? Please state it shortly in the revised version of the MS.

 Response: Thank you for your valuable comments. As you rightly pointed out, Sulcia and Nasuia are primary endosymbionts of the leafhopper Recilia dorsalis. During their long evolutionary process, primary endosymbionts have established close symbiotic relationships with their host insects. However, over the course of extended coevolution with the host, they have lost many essential genes for independent survival outside the host insect and typically reside within specific cells or tissues inside the host.

In our study, we obtained bacteriocytes from the leafhopper, which is a specialized tissue where Sulcia and Nasuia reside. By extracting DNA from bacteriocytes, we aimed to minimize contamination from the host insect Recilia dorsalis' genomic DNA and obtain a higher proportion of genomic DNA specifically from Sulcia and Nasuia. This approach not only reduces contamination but also ensures a more accurate representation of the genomes of these primary endosymbionts.

In contrast, in the study conducted by Kaszyca-Taszakowska & Depa (2022) titled "Microbiome of the Aphid Genus Dysaphis Börner (Hemiptera: Aphidinae) and Its Relation to Ant Attendance," they used high-throughput sequencing to obtain 16S rRNA information of all microorganisms present in the ant's body, representing the microbiome. Therefore, in their research, they used whole-insect DNA extraction.

Once again, we sincerely appreciate your insightful comments, which have led us to improve the methodology and interpretation in our study. Your input has significantly contributed to the rigor and accuracy of our research.

  1. at line 321 there is 2.8 billion yers which is absolutely impossible. Please correct it and provide correct age of this symbiosis.

 Response:

Thank you for your reminder. We have made the necessary correction to this error in the revised manuscript. According to our revised version, approximately 260 to 280 million years ago, the ancestors of Auchenorrhyncha established a symbiotic relationship with Sulcia and betaproteobacterial symbionts. For specific details, please refer to the revised manuscript, specifically lines 332-334. Your feedback has been invaluable in improving the accuracy of our study, and we truly appreciate your diligent review.